# Raman Research on Bleomycin-Induced DNA Strand Breaks and Repair Processes in Living Cells

**DOI:** 10.3390/ijms23073524

**Published:** 2022-03-24

**Authors:** Michał Czaja, Katarzyna Skirlińska-Nosek, Olga Adamczyk, Kamila Sofińska, Natalia Wilkosz, Zenon Rajfur, Marek Szymoński, Ewelina Lipiec

**Affiliations:** M. Smoluchowski Institute of Physics, Jagiellonian University, Łojasiewicza 11, 30-348 Kraków, Poland; michal.czaja@doctoral.uj.edu.pl (M.C.); katarzyna.skirlinska@gmail.com (K.S.-N.); olga.adamczyk@doctoral.uj.edu.pl (O.A.); kamila.sofinska@uj.edu.pl (K.S.); natalia.szydlowska@uj.edu.pl (N.W.); zenon.rajfur@uj.edu.pl (Z.R.); ufszymon@cyf-kr.edu.pl (M.S.)

**Keywords:** Raman spectroscopy, hyperspectral mapping, bleomycin, cell, HeLa, DNA damage, DNA repair, fluorescence microscopy, multivariate data analysis, PCA, HCA, NMF

## Abstract

Even several thousands of DNA lesions are induced in one cell within one day. DNA damage may lead to mutations, formation of chromosomal aberrations, or cellular death. A particularly cytotoxic type of DNA damage is single- and double-strand breaks (SSBs and DSBs, respectively). In this work, we followed DNA conformational transitions induced by the disruption of DNA backbone. Conformational changes of chromatin in living cells were induced by a bleomycin (BLM), an anticancer drug, which generates SSBs and DSBs. Raman micro-spectroscopy enabled to observe chemical changes at the level of single cell and to collect hyperspectral images of molecular structure and composition with sub-micrometer resolution. We applied multivariate data analysis methods to extract key information from registered data, particularly to probe DNA conformational changes. Applied methodology enabled to track conformational transition from B-DNA to A-DNA upon cellular response to BLM treatment. Additionally, increased expression of proteins within the cell nucleus resulting from the activation of repair processes was demonstrated. The ongoing DNA repair process under the BLM action was also confirmed with confocal laser scanning fluorescent microscopy.

## 1. Introduction

In every single cell, even several tens of thousands of DNA lesions occur each day [1]. Several specific types of DNA damage may be distinguished: point mutations, spontaneous deaminations, missing bases, formation of pyrimidine dimers, single-strand breaks (SSBs) and double-strand breaks (DSBs). Both types of strand breaks are cytotoxic. However, DSBs are the most dangerous type of DNA damage, leading to genetic instability [2]. Numerous DSBs can stop the processes of replication and transcription, thereby causing mutations and genomic aberrations [3,4]. If left unresponsive, such changes may compromise cell function and even lead to apoptosis [1]. To prevent consequences of serious DNA damage, cells have developed specific defense processes called DNA damage response (DDR). DSBs can be repaired mainly through two mechanisms: non-homologous end-joining (NHEJ) and homologous recombination (HR) [5,6].

DNA lesions may appear spontaneously or can be induced by various physicochemical factors including ionizing radiation [4,7,8], free radicals [9], chemicals [4], and drugs such as chemotherapeutic agents [10,11]. An example of chemotherapeutic drug, which causes DNA backbone disruptions is bleomycin (BLM) [11]. BLM binds to DNA minor groove in a sequence-dependent manner (recognizing dinucleotides, mainly GC and GT) and causes the strand rupture [10,12,13]. Both single- and double- strand breaks are induced via BLM. The mechanism of DNA cleavage is not fully understood, and several models have been proposed [10]. SSBs can be caused by the reaction of the bleomycin molecule with a pyrimidine nucleotide in the DNA chain, after which bleomycin becomes inactive. SSBs may occur randomly in the close region at the opposite strands of double-stranded DNA, eventually leading to DSBs. It was also reported that one individual bleomycin molecule may lead to complete DNA cleavage. In this case, after the first cutting event the inactive bleomycin molecule must be restored to its active state before the induction of the second cutting event of the complementary strand [10].

DNA conformation plays a crucial role in DNA susceptibility to damage formation and repair [14]. Under physiological conditions, DNA occurs in its native conformation known as the B-DNA. Another biologically significant DNA conformation is A-like form. Both are right-handed helices. A-DNA has a larger diameter and is shorter of about 20% comparing to the B-like DNA form. Different dimensions of these DNA forms determine different size of major and minor grooves. Since DNA interacts with various macromolecules via groves, these two DNA forms differ in the ability to bind biomolecules (e.g., repair proteins) because of the different geometry of these two structures. It was demonstrated that damaged DNA changes its conformation from B-DNA form to A-DNA [14,15,16], which is known to promote the activation of repair processes and expression of proteins involved in these mechanisms [17,18]. The B-to-A-DNA conformational transition is postulated to be a local effect [14]. The research on the local nature of structural modifications of the DNA molecules are prevented or largely prohibited due to the methodological limitations and insufficient sensitivity of conventional spectroscopic methods. The development of a combination of highly sensitive spectroscopic techniques with multivariate data analysis can become an efficient tool to investigate molecular modifications of DNA structure exposed to damaging factors.

In this article we report on conformational transition of DNA in living cells resulting from pharmacologically induced strand breaks. DNA damage was induced in HeLa cells treated with various concentrations of bleomycin. Chemical changes occurring in the cell in response to drug treatment were detected with the Raman micro-spectroscopy. To reduce dimensionality of spectroscopic data and extract the most important information, we applied multivariate data analysis. Raman maps of individual HeLa cells were analyzed with Hierarchical Cluster Analysis (HCA), Non-negative Matrix Factorization (NMF), and Principal Components Analysis (PCA) algorithms. Additionally, for spectra attributed to cellular nuclei (after HCA clustering of Raman maps) we performed PCA to follow concentration-dependent and incubation time-dependent spectral changes induced via BLM. The ongoing repair processes in response to chemical treatment were confirmed with fluorescence imaging. Specific fluorescent markers were used to visualize the presence of H2A.X histone phosphorylation, which is a hallmark of genetic material repair (see Section 4) [19,20]. The applied experimental, analytical and statistical approach is presented schematically in Figure 1.

## 2. Results

### 2.1. Raman Imaging and Multivariate Data Analysis of Individual HeLa Cells

In the presented paper, we follow the response of HeLa cells to bleomycin treatment. Detection of molecular changes of DNA in living cells require high chemical selectivity, which is provided by Raman spectroscopy. This approach allows to carry out experiments with samples immersed in liquid, and thus, to study of living cells in their native environment [21]. Raman mapping was applied to observe concentration-dependent chemical changes under three concentrations of DSBs inducing drug: 50 µM, 150 µM and 500 µM of bleomycin (Figure 2 and Figure 3). The control involved the population of untreated cells. Moreover, we tested two different incubation times: 24- and 48-h treatment. Arrays of Raman spectra were collected from individual HeLa cells, creating hyperspectral Raman maps [21,22,23,24,25,26] (see Section 4).

Methods of vibrational spectroscopy, in particular Raman hyperspectral imaging, provide multidimensional data. Appropriate interpretation of the obtained data requires reduction of its dimensionality. Application of a multivariate data analysis ensures reduction of data dimensionality and what is more allows extraction of useful information regarding the structure of data set, spectral similarity and variability.

Sub-micrometric spatial resolution of acquired Raman maps enabled to distinguish cellular organelles such as nucleus and nucleolus and to separately analyze DNA damage occurring in these structures. HCA was applied in order to divide spectra acquired from the cell into groups, that could be classified as separate organelles. HCA [27] is a method of clustering based on the spectral similarity and variability determined in a greedy manner. The major advantage of using HCA is producing the false-color maps established by the dendrograms, that illustrate the arrangement of the clusters determined during the analysis [23,27]. HCA compares acquired spectra directly. Therefore, a spectral signal of each voxel indicates the presence of various functional groups, from different biocomponents for example phosphates from DNA backbone and phospholipids. Therefore, the application of HCA to follow molecular changes separately in nucleic acids, proteins or lipids is limited and this is the main disadvantage of this approach. HCA analysis of acquired hyperspectral maps divided spectra of each cell into four or five clusters, corresponding to various cellular organelles: nucleus (red), nucleolus (yellow), cellular membrane (green), cytoplasmic areas (blue, and cyan) (see example in Figure 2).

Hyperspectral maps of HeLa cells were independently analyzed with NMF [28] and PCA [29,30,31] algorithms. In the NMF, the input data matrix is decomposed into two lower rank non-negative matrices W and H [23]. The first mentioned consists of scores related to the individual chemical bases presented in the matrix of H factors. The overall mathematical approach as well as the purpose of reducing the dataset dimensionality behind this method is similar to PCA. However, the main benefit of using the NMF analysis is the possibility to identify and differentiate biochemical compounds in the spectra with reasonable confidence based on the H matrix. It also allows producing the maps based on the scores (W matrix) that present the occurrence region of the component [26,28].

In this study, NMF analysis was applied to discover the sparse and meaningful features from the data, characteristic for each map component, including nucleic acids, lipids, proteins, and water. In comparison to HCA, NMF demonstrates main components of the Raman spectra from collected maps. Each of the presented components corresponds to different chemical composition present in particular regions of the cell (for example nucleic acids, water, proteins, etc... or sometimes their combinations). In contrast to HCA, NMF allows following spectral changes easier, for example in DNA, by direct comparison of DNA-related components of Raman maps acquired from various cellular samples. We presented three main components of spectra acquired from each individual HeLa cell, and visualized as NMF maps. The first component corresponds to cellular nucleus, while the second one is related to lipid-rich cytoplasmic structure. The last component visualized the distribution of O-H bending at 1643 cm^−1^, which corresponds to water distribution.

HCA and NMF results are presented in Figure 2, Figure 3 and Figure 4. Specifically, the results for a control group are presented in Figure 2. Bleomycin-treated cells undergoing repair processes and in an early apoptotic stage are shown in Figure 3 and Figure 4, respectively.

The analysis of spectral maps was additionally performed with PCA. The PCA is a linear transformation method that outlines the data into a new orthogonal space described by the axes of Principal Components (PCs) [23,29,30,31]. In order to obtain the predictive model (PCA model), the PCs, scores, and loadings values are computed. The aforementioned are the most significant for further conclusions. The scores represent the data in multidimensional space of the new variables called PCs. Loadings identify the variables that caused the separation noticeable in the scores matrix. These variables are recognized based on the position of extrema (both minima and maxima) in the loadings plot and compared with the scores. This method enables to determine the percentage of the total variance explained by each PCs of the original data set, which is useful for assessing the ability to reduce the dimensionality of data during this exploratory analysis [23,29,30,31].

PCA enabled highlighting spectral differences between various regions of individual cells. PCA focuses on variables with strong correlation allow grouping both positively correlated and negatively correlated components together. On the other hand, NMF finds patterns with the same direction of correlation. Therefore, NMF allows efficient extraction of components associated to individual biomolecules while PCA visualizes rather their combinations by distributions of positively and negatively correlated loadings [32].

For each cell we decided to present three main PCA components of the acquired Raman images. The results are presented in the Appendix A. For each analyzed map, first principal component was related to intensities of bands corresponding to proteins and nucleic acids. Therefore, the false color map of the distribution of the first PC visualizes cytoplasmatic and nuclear region of the cell (Appendix A). The second (control group) and the third (cells treated with bleomycin) PCs are positively correlated with bands attributed to water. The positive correlation with the methyl and methylene bending at 1445 cm^−1^, is visible in the 3rd loading plot for control and 2nd loading plot for bleomycin treated cells (Appendix A). These loadings indicate negative correlation with bands attributed to nucleic acids.

### 2.2. The Analysis of Bleomycin-Induced Changes in the DNA Backbone

A detailed analysis of bleomycin-induced changes in the DNA backbone was also performed. To avoid baseline influence and clearly demonstrate bleomycin-induced conformational changes, second derivatives of the averaged Raman spectra from HCA clusters attributed to cell nuclei were calculated and compared. A comparison of the second derivatives of averaged spectra acquired from control group and cells treated with various concentrations of BLM is presented in Figure 5a. The zoomed area of the ν_s_(PO_2_^−^) is demonstrated since the spectral position of this band is one of the main spectral markers of the DNA conformation. Moreover, second derivatives of NMF loadings are presented in Figure 5b. The zoomed area demonstrates the spectral position of phosphate symmetric stretching.

### 2.3. Fluorescence Imaging of Fixed HeLa Cells

Fluorescence imaging was performed to confirm repairing processes of chromatin in cell nucleus after the induction of DSBs with bleomycin. To visualize chromatin (in the whole areas of cellular nuclei) DAPI staining was applied. The applied fluorescence dye binds to phosphorylated H2A.X histone, which allows a direct visualization of DNA repair focal points [19,20]. Cells were incubated with bleomycin under analogue conditions, therefore, three bleomycin concentrations and the control group were involved, as well as measurements for two different incubation times were performed.

To achieve the highest possible spatial resolution, the experiment was performed using the confocal laser scanning microscopy (CLSM) system. Examples of fluorescent images of stained fixed HeLa cells are presented in Figure 6.

### 2.4. Multivariate Data Analysis of HeLa Cell Population Incubated with a Bleomycin

In this part of the work, the Principal Component Analysis (PCA) was performed as well to determine the effect of bleomycin on the cellular genetic material. In order to investigate effect of bleomycin on cellular chromatin, several models of PCA were performed on Raman spectra from the nuclei of at least three cells from different samples. These spectra were separated from the hyperspectral images through the previously used hierarchical cluster analysis (HCA). To visualize spectral changes resulting from the application of various bleomycin concentrations and different incubation times, several models of PCA were applied. For monitoring a concentration-dependent effect of BLM on cells, we analyzed spectra acquired from the cells treated with various concentrations of BLM with a single selected incubation time: 24 or 48 h. Additionally, to follow incubation time-dependent effect, we applied three other PCA models, each one for the selected BLM concentration. The results demonstrating the dose-dependent analysis (3D scores plot and loading plots) are shown in Figure 7. The incubation time-dependent effect is presented for each BLM concentration: Figure 8—50 µM of bleomycin, Figure 9—150 µM of bleomycin, Figure 10—500 µM of bleomycin. In general, for each calculated PCA model, three principal components are presented. For the sake of clarity all 2D projections of the score plots are presented in Appendix A.

## 3. Discussion

The applied data analysis, specifically HCA, allowed to demonstrate the spatial location of main cellular compounds including nucleus, nucleolus, cytoplasm, and membrane edge (see Figure 2b,c). The mean Raman spectra of each cluster were calculated to provide information about chemical composition of the cellular components. Spectra acquired from the cellular membrane (green cluster) and part of the cytoplasm (blue cluster) indicated relatively week signal from characteristic bands of proteins such as the ring breathing modes from phenylalanine rings at 1008 cm^−1^ [33,34], amide III in the spectral range from 1220 cm^−1^ to 1350 cm^−1^ [35,36], or δ(CH_2_, CH_3_) at 1450 cm^−1^ [33,35,37] due to lower concentration of these biomolecules than in central part of the cell. Raman spectra acquired from the remaining part of the cytoplasm (cyan cluster) are characterized by relatively high intensity of the δ(CH_2_, CH_3_) band, in comparing to the spectra extracted from the nucleus region. In addition, the Phe band had similar intensity in the spectra of cellular nuclei and some cytoplasmic areas (cyan). This observation indicates high concentration of lipids and proteins suggesting the presence of lipid droplets [38] or some cellular organelles such as Golgi apparatus and endoplasmic reticulum [39] that are present there. Relatively high intensity of nucleic acid bands such as the phosphate symmetric stretching ν_s_(PO_2_^−^) at 1098 cm^−1^ [33] and the vibration of guanine at 1343 cm^−1^ [40] were observed in the spectra acquired from cellular nuclei (red). The protein bands including the amide III and the Phe ring breathing mode are also well resolved in the spectra of nuclei areas. Distribution of intensities of selected characteristic Raman bands within HeLa cell are presented in Appendix A.

HCA performed on the Raman hyperspectral map of the cell treated with 150 µM BLM allowed nucleolus (yellow cluster) to be spatially localized. The reason of that may be the cell was stopped at G1 phase, due to G1/S DNA damage checkpoint [41]. High protein expression and genetic material duplication characteristic for cells being in G1/S phase of cell cycle may affect spectra improving signal-to-noise ratio (see Figure 3a). The Raman spectra of nucleolus and nucleus are similar, however intensities of some bands including the amide III, Phe, ν_s_(PO_2_^−^) vibrations are higher in the spectra acquired from nucleoli. In case of bleomycin-treated cell, intensities of Phe and amide III bands from the whole nuclei region are enhanced in comparison to the bands from cytoplasm. This indicates high protein expression in nuclei of cells treated with BLM. The high protein content is likely related to synthesis of enzymes, participating in the ongoing repair of DNA damage [16].

The separation of cellular organelles is not clearly visible at the typical map acquired from the early apoptotic, spheroidal cell, which was incubated with 500 µM bleomycin for 24 h (Figure 4). This is due to the loss of the adhesion, water-loss, and shrinkage—a hallmarks of cells undergoing apoptosis [42,43]. The high intensity of the DNA backbone bands in the spectra collected from the central area of the cell is worth noticing. This can be explained by relatively high density of cellular material in this area, due to the mentioned cell size reduction and its dehydration. Moreover, for an apoptotic cell, relatively high intensity of the CH_2_, CH_3_ bands in the central region was observed. This might indicate condensation of lipid droplets and formation of apoptotic bodies [16,44].

The second statistical approach to spectral maps consists of NMF analysis. The first NMF component, which corresponds to nucleus demonstrates relatively high (in comparison to other components) intensity of the bands from nucleic acids such as the symmetric stretching of phosphate at 1098 cm^−1^ [33,34]. The second component was related to cytoplasmic regions of cells with higher intensity of bands from lipids including CH_2_ and CH_3_ deformation at 1450 cm^−1^ [33,35,37]. Therefore, we assumed that this component corresponds to some lipid-rich cellular organelles. In contrast to HCA, NMF demonstrates the spectral difference between nucleus and cytoplasmic components, also for apoptotic cells. The last demonstrated NMF component was related to liquid medium. Therefore, only the wide band from the OH bending motion in water at 1643 cm^−1^ was dominant [45].

The analysis of bleomycin-induced changes in the DNA backbone (Figure 5) enabled to observe the phosphate band at 1099 cm^−1^ in the spectra of control cells. This spectral position indicates B-DNA conformation [46]. In the spectra collected from 150 µM BLM-treated cells a slight shift of this band to 1101 cm^−1^ was observed, indicating the partial conformational change from B-DNA to A-DNA. This conformational change is a hallmark of DNA repair processes [14,15,16,46,47]. However, in the averaged spectra the observed peaks from phosphate groups are derived not only from DNA but also other biomolecules such as phospholipids. Therefore, detection of the DNA conformational change through the shift of phosphate bands in such spectra is difficult.

A comparison of second derivatives of NMF loadings corresponding to nucleic acids is presented in Figure 5b. Here, shifts related to conformational transitions of DNA are resolved better than bands in the spectra extracted from HCA analysis. In particular, a continuous shift from 1098 cm^−1^ for control cells to 1100 cm^−1^ for 150 µM BLM-treated cell and 1102 cm^−1^ for apoptotic cell were observed. This result indicates the conformational change from B-DNA to A-DNA, due to cellular repair processes [14,15,16,46,47]. NMF loading comparison is presented in Figure 5b.

Fluorescence microscopy imaging confirmed activation of DNA repair process induced by BLM treatment, which was not detected in control cells. Imaging of cells incubated with the lowest studied concentration of bleomycin (50 µM) enabled to observe a fluorescence from phosphorylated histones, visible as foci (Figure 6). Similar results were observed for cells treated with 150 µM of bleomycin. However, for this BLM concentration more foci were observed. For the highest applied bleomycin concentration (500 µM), the signal from histone phosphorylation was observed in the entire volume of cellular nuclei. It is worth noting, that for this concentration numerous cells were in the apoptotic stage. In consequence, apoptotic cells in the late-stage were removed from the sample surface due to multiple washing required by the applied protocol of fluorescence staining (see Section 4). Therefore, we observed cells in early apoptotic stage confirmed by morphological changes (shrinkage without evident cellular fragmentation). However, traces of DAPI blue fluorescence were observed outside the nuclei area, which might be the effect of membrane blebbing—a hallmark of early apoptosis [48,49,50].

### 3.1. Concentration-Dependent Effect of Bleomycin

PCA was performed on the population of Raman spectra acquired from cellular nuclei area to provide chemical changes related to bleomycin-induced DNA damage. Figure 7 demonstrates a comparison of two PCA models visualizing the spectra variability depending on the BLM concentration, separately for 24 and 48 h. The clear separation along PC1 of spectra incubated for 24 h can be observed. Spectra extracted from control cells and treated with the 50 and 150 µM of bleomycin are located mainly on the negative site of PC1 in contrast to spectra acquired from cells incubated with 500 µM of BLM, that are localized on the positive site of the PC1. The corresponding loading plot indicates that for the separation along the PC1 results mainly from protein bands, including the bending of CH_2_, CH_3_ at 1450 cm^−^^1^ [33,35,37], as well as the amide III at 1315 cm^−1^ [35,36] and the phenylalanine at 1008 cm^−1^ [33,34]. The high intensity of these bands is characteristic for the spectra located at the positive site of PC1. This might indicate changes in proteins expression resulting from induced DNA damage and repair. Overexpression of protein with a predominance of one conformation was also reported in study of response of prostate cancer cells exposed to X-ray radiation [51]. However, in that research increasing concentration of α-helix proteins was observed.

PC1 also indicates a change of the spectral shape in the range of the amide I as presented in Figure 7. Minimum of PC1 loading is observed at 1632 cm^−1^, which we attribute to random coils [52]. This band is characteristic for the control group and spectra acquired for cells incubated with 50 and 150 µM of BLM. While the band at 1678 cm^−1^ attributed to β-sheets and β-turns [52] is more intense in the spectra acquired from cells incubated with the highest applied BLM concentration. These spectral changes confirm modifications of the secondary structure of proteins. Similar observation was reported for cholangiocarcinoma cells during induced apoptosis [53].

It is worth noting, that the spectral region from 1550 cm^−1^ to 1750 cm^−1^ corresponded to vibrations of water molecules [45]. Therefore, a negative of the band at 1632 cm^−1^ may indicate a loss of water within the cell nuclei due to ongoing apoptosis [42,43].

The first principal component distinguished the recorded spectra mainly due to the changes in protein concentration within the nucleus. The separation along PC2 was also observed. Significant part of the entire population of spectra from bleomycin-treated cells are located on the negative side of the PC2 as can be seen in Figure 7. Only all spectra acquired from the control group are located on the positive value part of PC2. Other bands attributed to guanine deformational motions at 1494 cm^−1^ and 1344 cm^−1^, as well the band at 1445 cm^−1^ related to deformation of thymine, are dominative in loadings of the second principal component [54]. This suggests changes in the DNA molecular structure induced by bleomycin treatment. A minimum of the PC2 loading was observed for 1104 cm^−1^, which corresponds to ν_s_(PO_2_^−^) in A-DNA [15,16,46]. Therefore, we assumed that the spectra displaying negative value of PC2 were acquired from chromatin subjected to conformational transitions caused by ongoing repair processes. The ν_s_(PO_2_^−^) band was observed in control group at 1085 cm^−1^ indicating B conformation of DNA, as expected, please see loading plot corresponding to PC2 presented in Figure 7.

Clustering along PC3 was less evident for spectra acquires from cells after 24-h bleomycin-treatment. Only spectra from the control group were well separated. The loading plot of PC3, similarly to PC1, indicated the separation of spectra due to differences in bands from proteins, including amide III band at 1340 and 1270 cm^−1^, amide I at 1661 cm^−1^, and Phe at 1004 cm^−1^ [36].

Similar PCA results as those obtained for the spectra acquired from cells incubated with BLM for 24 h were also obtained for the second applied incubation time—48 h. The shape of all three loading plots displays a relatively high level of similarity confirming that the same bands are responsible for the clustering visible on the score plots. However, score plots obtained for these two models are slightly different. In both models, the cluster of control spectra is well separated. However, no clear separation along PC1 for spectra acquired from bleomycin-treated cells was observed. The increase of incubation time to 48 h did not result in additional biochemical changes of proteins upon BLM treatment.

The loading plot of PC2, which explains that the separation along PC2 is mainly related to spectral position of phosphate bands from DNA backbone and also vibrations of DNA bases including guanine and thymine. In the spectra of BLM-treated cells, that are located at the negative side of PC2 the ν_s_(PO_2_^−^) band is at 1104 cm^−1^ indicating A-DNA conformation [15,16,46]. All the spectra acquired from the control group of cells and part of population of spectra from BLM-treated cells are at the positive side of PC2. The band ν_s_(PO_2_^−^) in these spectra is observed at 1085 cm^−1^ confirming B-DNA conformation. The observed shift of the phosphate bands indicated partial conformational transition from B- to A-DNA in the cells treated with all of the applied doses of BLM.

### 3.2. Time-Dependent Effect of Bleomycin

The three other models of PCA were applied to follow time-dependent molecular changes for each applied BLM concentrations separately (Figure 8, Figure 9 and Figure 10). A clear separation of spectra along PC1 is observed in score plot presented in Figure 8, showing the time-depended influence of 50 µM of BLM. The loading plot of PC1 proves that the observed separation is related to protein bands (bending of CH_2_, CH_3_ at 1450 cm^−1^ [33,35,37], amide III at 1315 cm^−1^ [35,36] and Phe at 1008 cm^−1^ [33,34]), confirming high protein expression in cells treated with BLM due to the DNA repair. Similar spectral changes were described by Roman et al. for prostate cancer cells exposed to X-ray radiation [51]. Again, a part of the spectra population acquired from bleomycin-treated cells is located at the negative side of PC2. The corresponding loading plot of this main component indicates a conformational change from B-DNA to A-DNA as observed before. Additionally, clear separation between the spectra from the 50 µM group after 48 h and the rest (control group and 50 µM group after 24 h) was observed along the PC3. Loadings of this component indicate the influence of the protein bands.

PCA calculated for the control group and the concentration of 150 µM gave very similar results as the previous model investigating an influence of 50 µM BLM on HeLa cells. However, looking at the separation along PC1 and PC3, one may notice that the spectra acquired from cells incubated for 24 h are partially clustered with the control spectra and partially with the spectra of cells incubated for 48 h. This suggests that the DNA repair process does not take place in all cells simultaneously. According to the literature, several factors may influence the time scale of DNA repair process, including phase of cell cycle [55] and level of chromatin condensation [56]. All the control spectra and a part of spectra acquired from cells incubated with both applied concentrations of BLM are clustered on the positive site of PC2, in contrast to some spectra from treated cells, that are located opposite. As in previous models PC2 is related to DNA molecular structure, in particular its conformation. The separation of spectra along PC2 confirm the already mentioned partial conformational transition from B- to A-DNA form. This transition which is more pronounced after 48 h incubation with BLM than after 24 h—more spectra from the cells incubated for 48 h with BLM have negative values of PC2.

The last PCA model was performed for control spectra and spectra acquired from the nuclei area of cells treated with 500 mM of BLM. The separation along each PCs was more evident. This suggests significant time-dependent biochemical changes induced upon cellular response to the applied concentration of the drug. Similar to the results of previous models loading plots of PC1 and PC3 indicate that separation along these components is related to changes of the molecular structure of protein and peptides and their expression. PC2 loading plot indicates that the observed clustering of spectra is also associated with differences in DNA conformation, that is expected due to apoptosis, as observed elsewhere [16,51].

## 4. Materials and Methods

### 4.1. Cell Culture

In this study the immortalized cell line HeLa (American Type Culture Collection company, Manassas, VA, USA) was used. Cells were cultured in Dulbecco’s Modified Eagle’s Medium–high glucose (11965-084, Gibco, Waltham, MA, USA) buffer with addition of 5% BSA and 1% antibiotics (penicillin and streptomycin). Incubation was carried out at 37 °C with 5% CO_2_.

Cells were subcultured using the following procedure. In the first step, cultured cells were washed with DMEM buffer (without FBS and antibiotics), and next cells were treated with 0.25% trypsin with EDTA for 3 to 5 min until all cells were detached. Then, trypsin in a cell solution was deactivated with adding full DMEM HG medium (with additions).

Samples for Raman micro-spectroscopy experiment were prepared by seeding cells on silicon substrate with metallic coating (gold and titanium), previously covered by poly-L-lysine. Cells were incubated with bleomycin for 24 or 48 h. At least 3 cells were measured for each parameter set (BLM concentration, incubation time).

Prior to the fluorescence imaging experiment, cells were plated in a 35 mm glass bottom dishes with thickness 0.16–0.19 mm designed for high resolution imaging (Cellvis) and incubated with a bleomycin under analogous conditions (bleomycin concentration and incubation time) as in experiments involving Raman imaging. Cells were fixed with 4% formaldehyde solutions (47608, Sigma Aldrich, Saint Louis, MO, USA) for 10 min, permeabilized for 10 min in 0.5% Triton X-100 (T9284, Sigma). Fixed cells were incubated in a PBS buffer with 3% BSA (84065-M, Sigma) for 45 min, then stained for anti-phospho Histone H2A.X (1:100 in 3% BSA PBS buffer; 05-636-AF488, Sigma) for 35 min. For nucleus staining, cells were additionally incubated with DAPI (1:50 in deionized water; D9542, Sigma) for 5 min.

### 4.2. Raman Micro-Spectroscopy

Measurements were performed using a Horiba LabRam confocal micro-spectroscopy system (Horiba France SAS Ltd., Palaiseau, France) equipped with a green laser (λ = 532 nm), electron-multiplying charge-coupled device (EM-CCD) cooled to −70 °C, and water immersion 60× 1.0 NA objective (Nikon Instruments Inc., Tokyo, Japan). During spectra acquisition, the cell culture was immersed in physiological saline solution. Spectra were acquired in the fingerprint spectral region (1900–800 cm^−1^) with the spectral resolution of 2 cm^−1^. An exposure time for each pixel was 6 s (2 accumulations with 3 s acquisition). Mapping step size in *x*-*y* directions was between 0.5–0.7 µm (dependent from cell size). Raman mapping with the confocal system allowed registration of spectra from well-defined volume at each pixel of the acquired maps. This allowed to create a hyperspectral images of cells [21,22]. Measurements were controlled by LabSpec 6 software. Each map consists of ~7000 spectra. From each individual cell ca 2000 spectra were extracted via HCA including ~500 from the cellular nucleus.

### 4.3. Confocal Fluorescent Microscopy

Fluorescent confocal images were acquired using Zeiss LSM 710 confocal module set on Zeiss Axio Observer.Z1 inverted microscope (Carl Zeiss Microscopy GmbH, Jena, Germany) using an oil immersion 40× 1.4 NA Plan-Apochromat objective. The microscope was driven by Zen Black software (ver. 8,1,0,484, ZEN 2012 SP1). DAPI and Alexa Fluor 488 dyes were excited with 405 nm diode laser and 488 Ar laser, respectively.

### 4.4. Bleomycin

The preparation protocol of bleomycin solution was adapted from [50].

To initiate cleavage events of DNA backbone, bleomycin needs to be transferred into so called “activated” bleomycin, which requires the presence of ion metal, usually Fe(II) or Cu(I), and oxygen. In our work, bleomycin sulfate (B3972, TCI Europe N.V., Zwijndrecht, Belgium) was activated by adding ammonium iron (III) sulfate dodecahydrate (F3629, Sigma Aldrich, Saint Louis, MO, USA). The mass ratio of bleomycin to iron salt was 1:1.1. The resulting solution was neutralized by adding 0.1 M NaOH. The concentration of the obtained bleomycin solution was 10 mM. For incubation with the cell culture, the prepared BLM solution was further diluted in DMEM to obtain the following BLM concentrations: 50 µM, 150 µM and 500 µM.

### 4.5. Data Analysis and Processing

Prior to multivariate statistical analysis, specifically Hierarchical Cluster Analysis (HCA), Principal Component Analysis (PCA) and Non-negative Matrix Factorization (NMF), the Raman spectra were baseline corrected (5th order of polynomial), smoothed (Savitzky-Golay Filter, 7 smoothing points, 3rd polynomial order) and normalized (Standard Normal Variate) using MATLAB R2019b software (MathWorks, Inc., Natick, MA, USA). When necessary, a cosmic ray removal procedure was applied. The analysis was performed in the spectral range of 1800–900 cm^−1^, unique for biological molecules.

Microscopic images analysis was performed using Fiji ImageJ environment [57,58].

## 5. Summary and Conclusions

In the presented paper, the cellular response to bleomycin-induced DNA strand breaks in living HeLa cells was investigated. Raman micro-spectroscopic mapping of individual cells was applied to follow molecular changes induced upon the cellular response to bleomycin treatment (three different concentrations, 24- and 48-h incubation). This enabled us to identify spectral markers of DNA damage repair and apoptosis. The dimensionality of the obtained hyperspectral maps was reduced using several methods of multivariate data analysis. An application the two independent experimental approaches such as molecular spectroscopic mapping and fluorescent microscopy supported with multivariate data analysis enabled to achieve a complementary overview of molecular modifications in cells responding to bleomycin treatment. The applied comprehensive approach was confirmed to be an efficient tool in research on molecular modifications of the DNA structure exposed to the influence of damaging factor. This approach could be applied in further studies on structural modifications of DNA or other cellular components under the action of various external factors.

HCA allowed to extract spectra from cellular organelles including nuclei and analyze molecular changes, induced in chromatin upon DNA repair and apoptosis. For the each acquired spectral map we demonstrated three main components of NMF and PCA: related to the area of the nucleus, cytoplasm and surrounding medium. Comparison of the NMF components plots corresponding to cell nucleus showed blueshift of the ν_s_(PO_2_^−^), which indicates the change in DNA conformation.

To observe systematic changes caused by bleomycin treatment, PCA was performed on the population of Raman spectra distinguished by HCA. Several groups of spectra were selected and compared to separately follow spectral changes dependent on bleomycin concentration and incubation time.

The application of the three various statistical approaches allowed to compare and discuss their advantages and limitations in analysis of hyperspectral Raman maps. NMF provided the most straightforward visualization of conformational transition induced by bleomycin. On the other hand, PCA classified spectra extracted by HCA and pointed marker bands of cellular response to bleomycin treatment.

A strong correlation was observed between the increasing concentration of bleomycin and protein expression in the nucleus area. This observation may be related to the activation of DNA repair processes. The conformational transition from B-DNA to A-DNA resulting from strand breaks (SSBs and DSBs) of DNA induced by bleomycin was observed after 24 and 48 h of treatment with 50, 150 and 500 µM of BLM. Moreover, the process of DNA restoration to its native form after extended incubation (48 h, 50 and 150 µM BLM) was demonstrated. The activation of repair processes was additionally confirmed by fluorescence microscopy. DNA repair foci were observed as histone gammaH2AX phosphorylation—a hallmark of DNA repair process. The number of DSBs being under repair increased significantly with bleomycin concentration.

Bleomycin treatment with the highest applied concentration led to cell apoptosis, which was detected as characteristic spectral changes indicating water loss, increase of cellular material density and formation of apoptotic bodies. Additionally, a hallmark of cell death, specifically chromatin fragmentation was observed with fluorescence imaging.

## Figures and Tables

**Figure 1 ijms-23-03524-f001:**
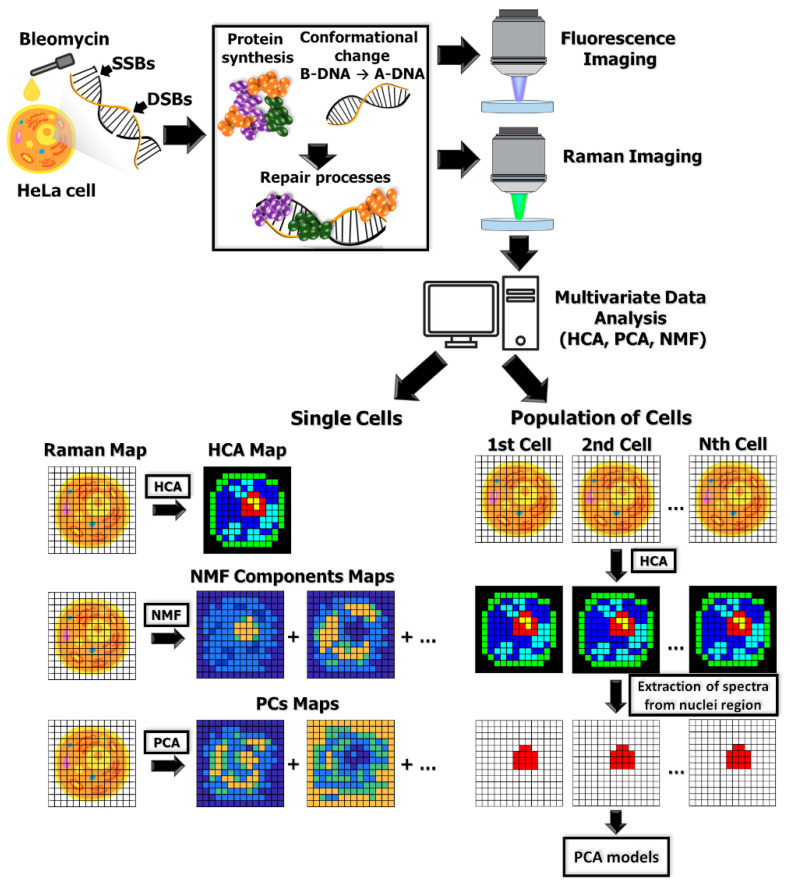
Schematic representation of experimental, analytical and statistical approach in the reported research. Bleomycin-induced DNA damage (SSBs and DSBs) leads to synthesis of repair proteins and DNA conformational change from B-DNA to A-DNA. Biochemical response to DNA damage induction can be observed as characteristic markers in Raman spectra or foci in fluorescence images. Raman images of individual cells were analyzed via HCA, NMF and PCA. As a result, false color maps were obtained. Spectra extracted using HCA algorithm from nucleus area of several control and BLM-treated cells were also analyzed with PCA.

**Figure 2 ijms-23-03524-f002:**
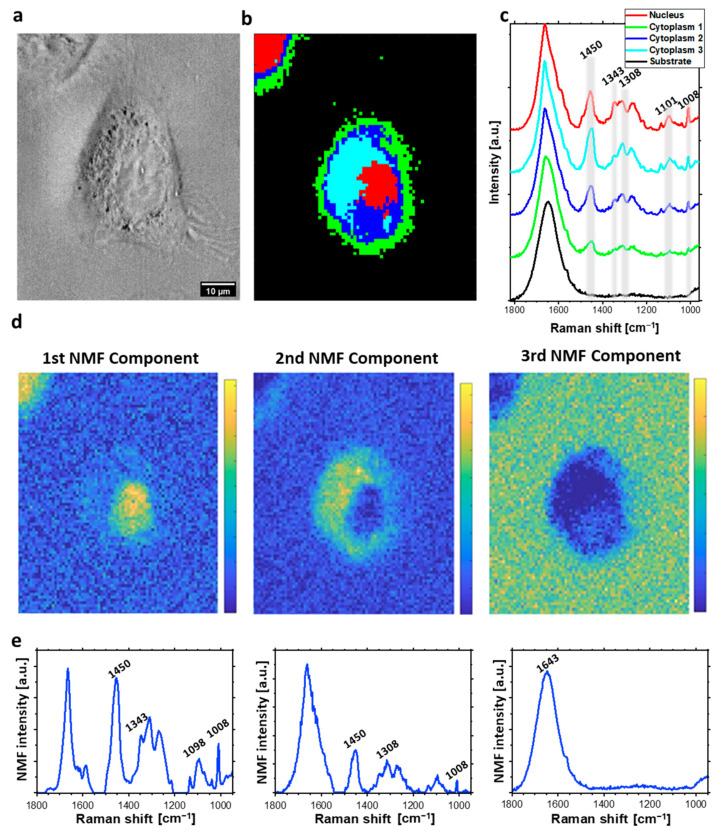
Raman imaging of control (untreated) HeLa cell: (**a**) The cell optical image; (**b**) HCA map; (**c**) HCA mean Raman spectra (color of each spectrum corresponds to color of the cluster in (**b**)); (**d**) distribution of NMF components; (**e**) corresponding plots of NMF components.

**Figure 3 ijms-23-03524-f003:**
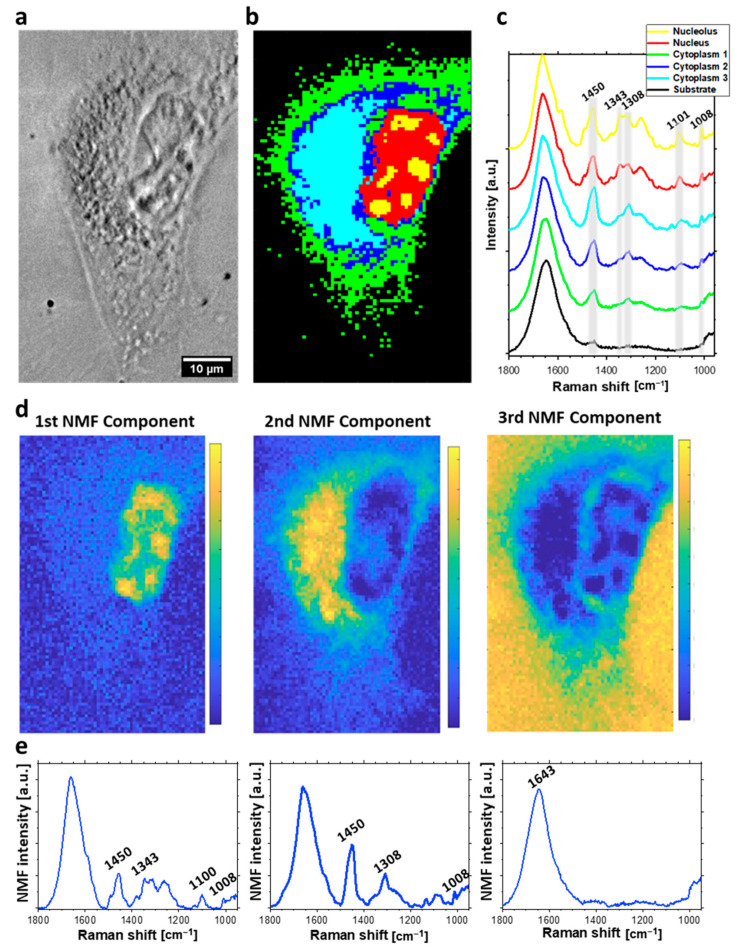
Raman imaging of HeLa cell incubated with 150 μM for 24 h: (**a**) optical image; (**b**) HCA map; (**c**) HCA mean Raman spectra (color of each spectrum corresponds to color of the cluster in (**b**)); (**d**) distribution of NMF components; (**e**) corresponding plots of NMF components.

**Figure 4 ijms-23-03524-f004:**
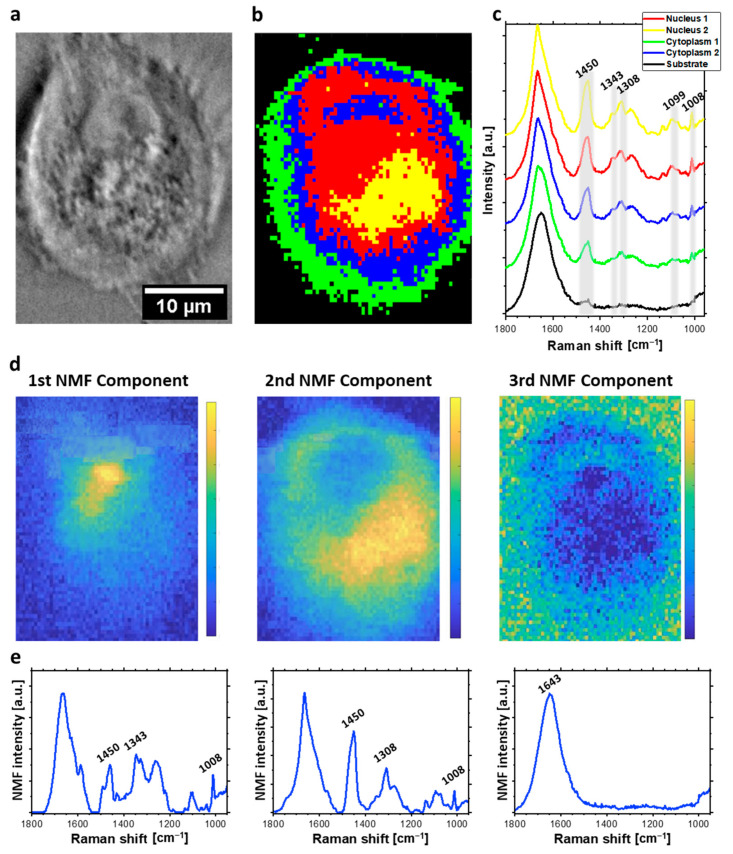
Raman imaging of apoptotic HeLa cell incubated with 500 μM for 24 h. (**a**) Cell optical image; (**b**) HCA map; (**c**) HCA mean Raman spectra (color of each spectrum corresponds to color of the cluster in (**a**)); (**d**) false-color image NMF maps of three components distribution; (**e**) corresponding plots of NMF components.

**Figure 5 ijms-23-03524-f005:**
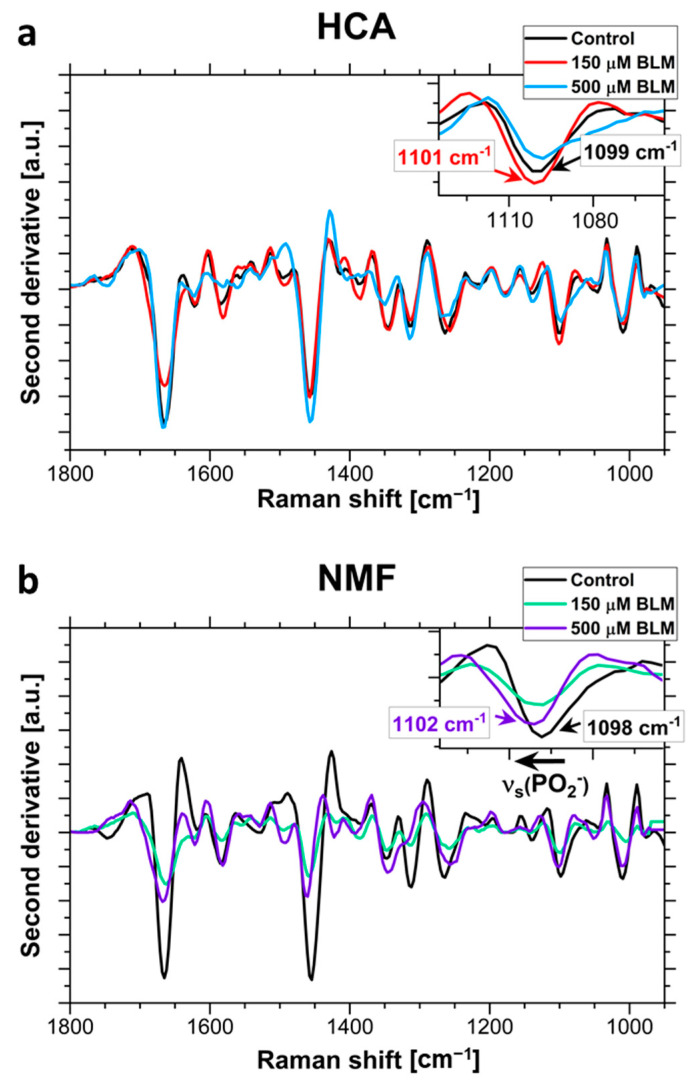
Monitoring of DNA conformational transitions in selected cell nuclei: (**a**) Second derivatives of Raman spectra extracted via HCA from nuclei region of HeLa cells. Zoomed spectral region attributed to νs(PO_2_^−^) is demonstrating shifts resulting from the conformational change from B-DNA to A-DNA. (**b**) Second derivatives of NMF components corresponding to signal from the nuclei area of HeLa cells. Zoomed spectral region attributed to νs(PO_2_^−^) is demonstrating shifts resulting from the partial conformational changes from B-DNA to A-DNA.

**Figure 6 ijms-23-03524-f006:**
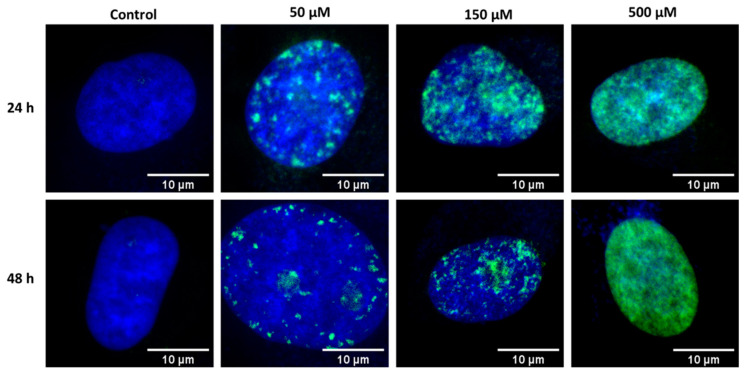
Fluoresce images of nuclei of individual fixed HeLa cells incubated with various bleomycin concentrations. From left to right side: control group, 50 µM of bleomycin, 150 µM, 500 µM. Top: 24 h of incubation; bottom: 48 h of incubation.

**Figure 7 ijms-23-03524-f007:**
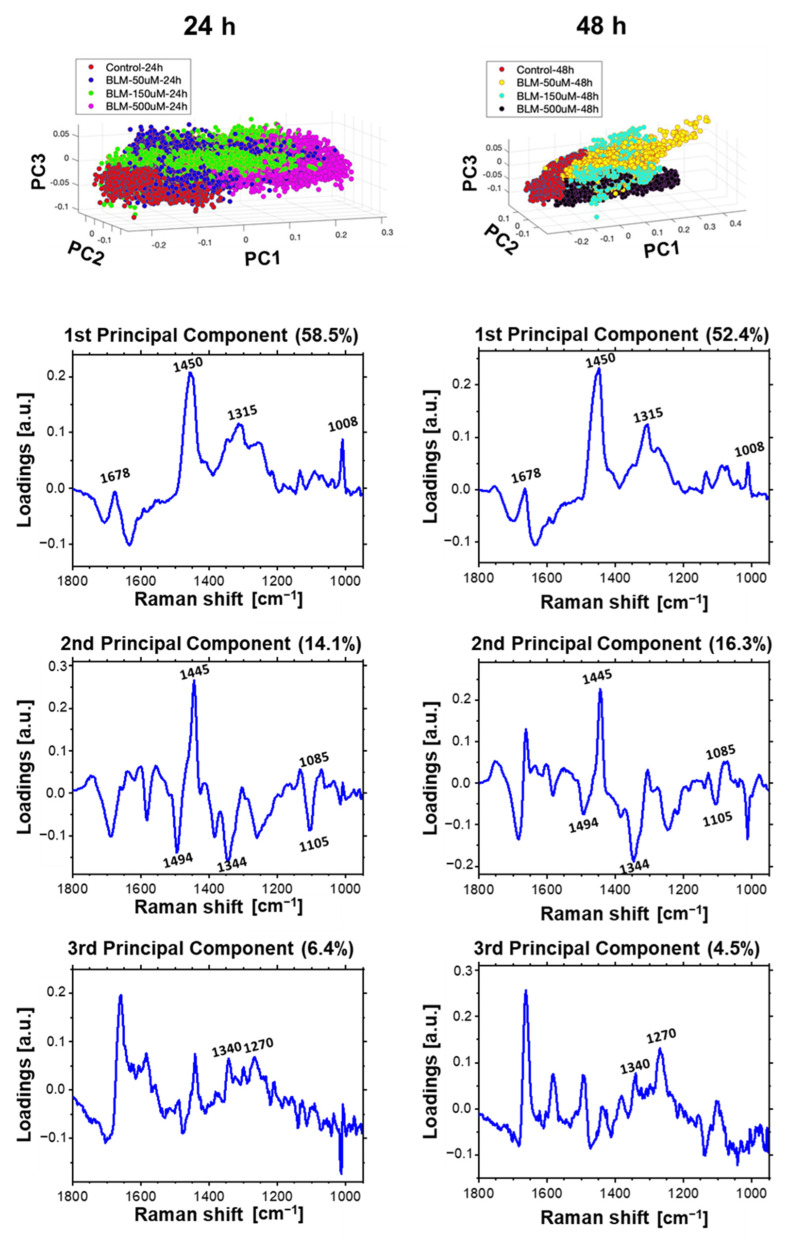
PCA of Raman spectra of cellular nuclei of HeLa cells incubated with bleomycin (control, 50 µM, 150 µM and 500 µM) for 24 (left 3D image) and 48 h (right 3D image). For each incubation time, three first most significant loadings are presented.

**Figure 8 ijms-23-03524-f008:**
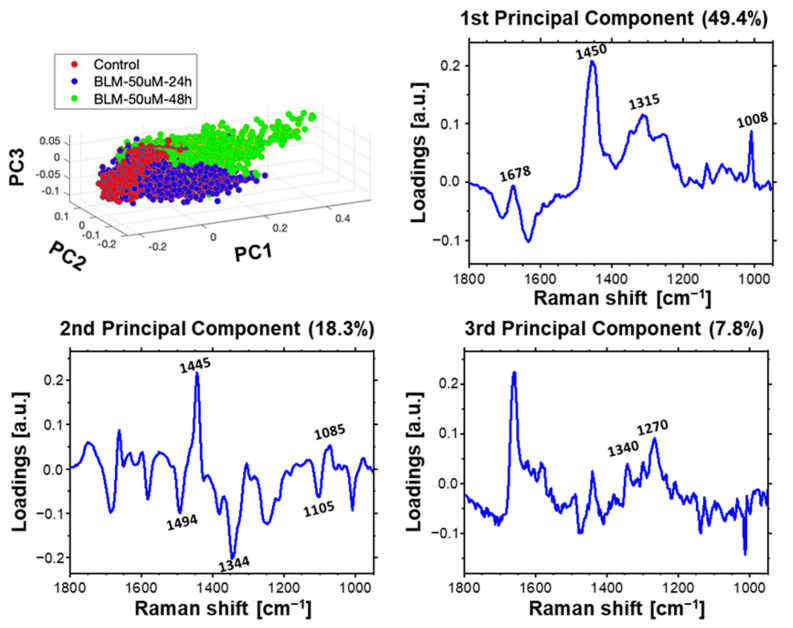
Time-dependent effect of BLM incubation. Results of PCA for Raman spectra acquired from nuclei of control cells and cells treated with 50 µM of BLM for 24 and 48 h. Loading plots of three first principal components are presented.

**Figure 9 ijms-23-03524-f009:**
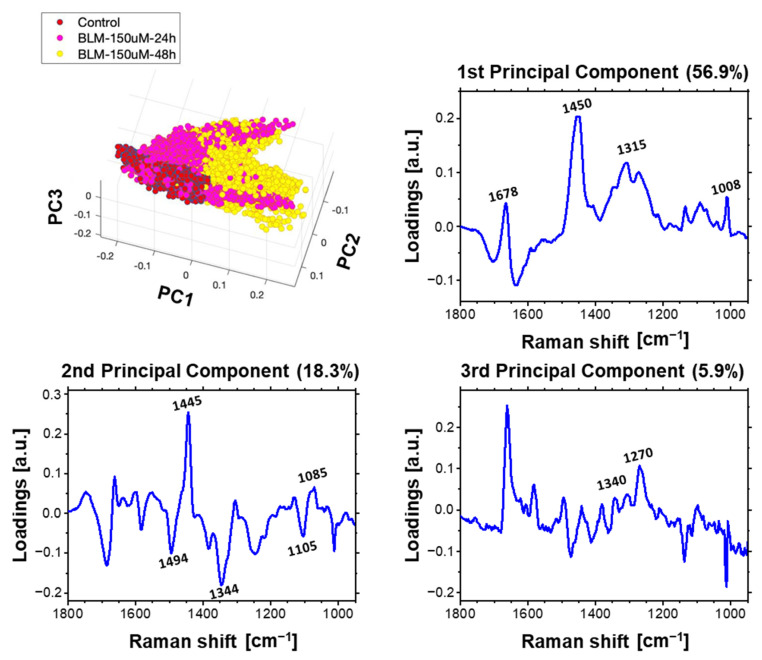
Time-dependent effect of BLM incubation. Results of PCA for Raman spectra acquired from nuclei of control cells and cells treated with. 150 µM of BLM for 24 and 48 h. Loading plots of three first principal components are presented.

**Figure 10 ijms-23-03524-f010:**
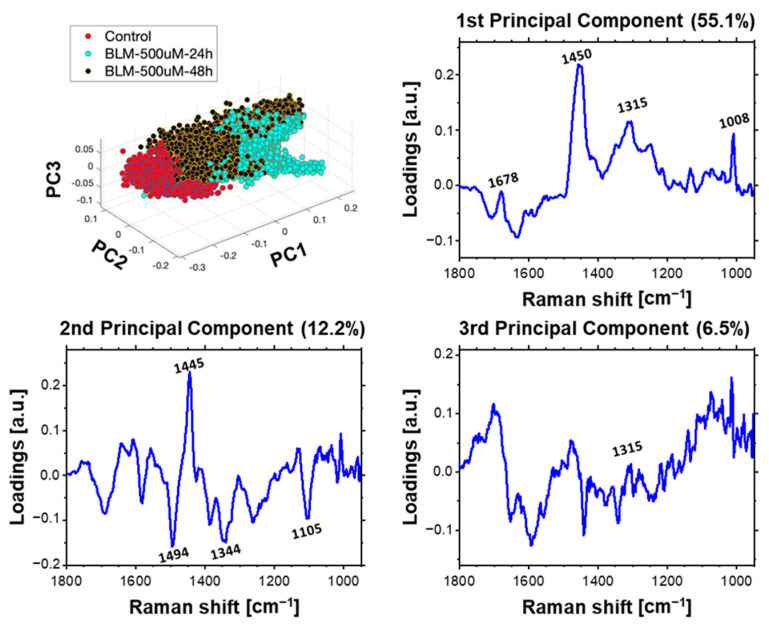
Time-dependent effect of BLM incubation. PCA results of Raman spectra acquired from nuclei of control cells and cells treated with. 500 µM of BLM for 24 and 48 h. Loading plots of three first principal components are presented.

## Data Availability

The data that support the findings of this study are available from the corresponding author upon reasonable request.

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
