# Peer review of "Raman Research on Bleomycin-Induced DNA Strand Breaks and Repair Processes in Living Cells"

_ijms, 2022, doi:10.3390/ijms23073524_

Round 1

Reviewer 1 Report

The manuscript entitled “Detection of bleomycin-induced DNA strand breaks and repair processes in living cells by Raman microscopy” by Czaja et, al. described a method of Raman microscopy. The experiments were well designed and performed. However, there are some fundamental issues to be carefully addressed before this manuscript get suitable for publication.

  1. The Raman imaging and florescence imaging are to analyze conformational change of DNA, indicating DNA strand break or repair processes. They are not direct detections of DNA strand breaks and repair processes. Thus, the title, as well as some descriptions in main text, might be somewhat overstate and misleading.
  2. The Introduction section is like a mini-review, which is not recommended. However, why analyzing DNA strand breaks and repair processes are important, what is known, what is the gap of knowledge/technique, still have not been well-described in Introduction.
  3. The discussion is even more problematic. Data analyses using HCA, PCA, NMF should all be moved to Results section. Detail description of how such data analyses where performed should be move to Methods section. Meanwhile, important information such as the advantage and disadvantage of each method (HCA, PCA, NMF), how the detection method introduced in this manuscript may move the field forward, etc. are still missing.

Reviewer 2 Report

This is a thorough and well communicated report on the application of sub-micrometer resolution Raman spectroscopy to monitor bleomycin-induced double-strand DNA breaks in cells.  I have only one small editorial suggestion.  The Introduction section contains quite a lot of details about the physical basis for Raman spectroscopy that might be better placed within the first part of the Discussion section.  By including this in the Introduction, the reader is distracted from the core purpose of the study, which is to assess the ability of sub-micrometer Raman spectroscopy to observe DNA damage and the associated processes resulting from bleomycin treatment.  I think a more streamlined Introduction that more quickly leads to experimental data accumulation and interpretation will make a more compelling story.  The physical details that underlie the techniques are better considered within the context of these experimental results than prior to any measurement.  This is just my opinion, but I believe a more concise Introduction leading more directly to observations upon applying the spectroscopic techniques to BLM-treated cells will improve the message of paper.

Reviewer 3 Report

This work present the application of Raman spectroscopy to study DNA break and repair processes in the nuclei of living cells, inducing the damage with bleomycin, a chemotherapeutic drug. Different methods of multivariate analysis are applied to understand the spectral differences of the data obtained from the cells under different treatments. The analysis and results are interesting, but several improvements need to be made.

In general, throughout the manuscript, references that connect certain statements or explanations to the corresponding figures are missing, or are written too far into the paragraph. This makes the manuscript harder to read and understand, specially in the discussion section. Also, the word "amide" is misspelled several times through the text.

1- In the Introduction, the description of the multivariate analysis methods that will be used is very detailed, but the pros and cons of the methods are not well emphasized. A clear comparison between them that would explain why all of them are used in this work would help with this point.

2- In the Introduction, the authors state that "the main benefit of using the NMF analysis is the possibility to identify and differentiate biochemical compounds in the spectra". However, similar information can be extracted from the other multivariate methods used, since for example the principal components shown from PCA also give information of biochemical compounds. Can the authors please clarify?

3- In the Results section, there is a mention that cells were analyzed with NMF and PCA (starting in line 164). In the end of that paragraph, the authors write "For both NMF and PCS, components validity is visualized as false-color images, and moreover, the NMF components and PC loading plots that demonstrate marker bands of cellular compounds are present". However, there is no reference to a figure where this information could be found. 

4- In Figure 6, it should be stated that the images correspond to the nucleus and no to the whole cell, or change the images to whole cells. Also, in the corresponding paragraph of Fluorescence imaging, it should be stated that DAPI is used, this would make Figure 6 easier to understand. 

5- In the Discussion section, line 276, the authors state that "These processes technically reduced the access of the Raman laser beam to various cellular compartments", referring to water-loss and shrinkage of the cells. Why is the "access of the Raman laser beam" reduced? Or do the authors mean that the focal volume in the cells is now more crowded and spectra are more difficult to interpret?

6- In line 292, references 40 and 41 are not associated with the spectral position of B-DNA conformation. Please provide the correct references.

7- In the discussion of the phosphate band shifts around 1100cm-1 (Figure 11), the phosphate bands observed for the spectra obtained from the HCA cluster related to the nuclei are said to be due to not only phosphate groups of DNA, but also from phospholipids. For the NMF component of the nuclei, the same phosphate bands are considered to be only from phosphate groups of DNA. Is there another band in the spectra or in the NMF component that indicates this distinction? If the spectra used for each analysis are from the same region, why is one considered to have contributions from several phosphate sources and not the other?

8- In the discussion of the concentration effect of bleomycin, some Raman shifts mentioned in the text are not marked in the loading plots. For example, the shift at 1632cm-1 is mentioned but not marked.

9- In that same discussion, line 373, the intensity of the 1675cm-1 band is discussed, but no reference of spectra is provided to see this statement. In line 392, the authors write that "the PO2- band was observed in control group at 1085cm-1 indicating B conformation of DNA", but no reference to spectra is provided to see this. And in the last paragraph, the band is discussed again but there is no spectra reference to see this. Please provide the correct references to see .

10- In the Materials and Methods section, it is mentioned that at least three cells for each parameter set were measured. Could the authors provide the average amount of spectra obtained for each cell, and the average amount of nuclei spectra separated by HCA? Also please provide more detailed information of the fluorescence measurements (lasers or filters used).

11- In line 520, the methods acronyms are repeated.

12- The Conclusions would benefit from a deeper discussion on how the multivariate methods used helped in the study of DNA damage, and a comparison between them to determine which one would be suited better for these applications or why all of them would be needed.

13- The PCA color maps are presented in the supplementary are only mentioned once and not discussed nor compared to the NMF and HCA results. Do they provide any new information that has not been obtained from the other methods?

Round 2

Reviewer 1 Report

I don't have further comments to this version.

Reviewer 3 Report

Thank you for your replies. With the provided corrections, I recommend the acceptance of this manuscript.